# Personal Identity and Uncertainty in the Everett Interpretation of Quantum Mechanics

Zhonghao Lu

Department of History and Philosophy of Science, University of Pittsburgh, Pittsburgh, PA 15260, USA; lu_zhonghao@pitt.edu

**Abstract:** The deterministic nature of EQM (the Everett Interpretation of Quantum Mechanics) seems to be inconsistent with the use of probability in EQM, giving rise to what is known as the "incoherence problem". In this paper, I explore approaches to solve the incoherence problem of EQM via pre-measurement uncertainty. Previous discussions on the validity of pre-measurement uncertainty have leaned heavily on intricate aspects of the theory of semantics and reference, the embrace of either four-dimensionalism or three-dimensionalism of personhood, or the ontology of EQM. In this paper, I argue that, regardless of the adoption of three-dimensionalism or four-dimensionalism of personhood, the overlapping view or the divergence view of the ontology of EQM, the pre-measurement uncertainty approach to the incoherence problem of EQM can only achive success while contradicting fundamental principles of physicalism. I also use the divergence view of EQM as an example to illustrate my analyses.

**Keywords:** Everett Interpretation of Quantum Mechanics; personal identity; probability in quantum mechanics

## 1. The Incoherence Problem

The Everett Interpretation of Quantum Mechanics (EQM) is a deterministic physical theory, but it also involves probability via the Born Rule. (See [1] for an overall introduction. In [2], Everett attempted to reconstruct the Born Rule in Section 5, while assuming full determinism as the underlying principle of Quantum Mechanics.) The deterministic nature of EQM seems to be inconsistent with the use of probability in EQM. This has been called the "incoherence problem" of EQM [3].

Consider the simplest branching process with only two branches. Imagine an observer, Aristotle, measuring the *z*-spin of an electron in a state of superposition of different *z*-spins. The initial state of the entire system is represented by $\frac{1}{\sqrt{2}}(|\uparrow\rangle + |\downarrow\rangle) \otimes |\text{Aristotle } 0\rangle$, where $\frac{1}{\sqrt{2}}(|\uparrow\rangle + |\downarrow\rangle)$ represents the initial state of the electron, and $|\text{Aristotle } 0\rangle$ is the initial state of Aristotle. After the measurement, the state of the whole system evolves into $\frac{1}{\sqrt{2}}|\uparrow\rangle \otimes |\text{Aristotle } \uparrow\rangle + \frac{1}{\sqrt{2}}|\downarrow\rangle \otimes |\text{Aristotle } \downarrow\rangle$, where $|\text{Aristotle } \uparrow\rangle$ (or $|\text{Aristotle } \downarrow\rangle$) signifies the state of Aristotle seeing the *z*-spin is up (or down). From an "outside" viewpoint, all branches equally exist after the measurement, and both the probabilities of Aristotle seeing the *z*-spin is up and Aristotle seeing the *z*-spin is down are 1. But from an "inside" viewpoint, one can only obtain one single result after the measurement. Consequently, according to the Born Rule, both the probabilities of Aristotle seeing the *z*-spin is up and seeing the *z*-spin is down are 1/2 [4].

In a deterministic theory, the following principle is commonly held true.

Ignorance: In order to make propositions such as "the probability that event *E* happens is *p*" meaningful in a deterministic universe, we must be ignorant of some facts about E.

*Ignorance* is commonly acknowledged in classical physics. In the background of classical mechanics as a deterministic physical theory, whether it will be raining tomorrow is determined by the physical state of a given moment, *s*. But we cannot discern which physical state it is among a vast array of similar physical states $\{s'\}$. This is the basis for discussions involving probability in classical mechanics. Loosely speaking, if the measure of all states $\{s'\}$ is *A*, and the measure of those states in $\{s'\}$ that lead to tomorrow's rain is *B*, then the probability that it will rain tomorrow is $B/A$ given the physical state is *s*. This probability arises from our ignorance of the precise physical state of this moment.

In this paper, I shall explore one line to solve the incoherence problem via pre-measurement uncertainty. I shall focus on Saunders and Wallace's proposal that some kind of pre-measurement uncertainty, which comes from the lack of specific *indexical knowledge* of observers, can resolve the incoherence problem in EQM [3,5–10]. According to Saunders and Wallace, even though Aristotle knew that the state of the entire system would be $\frac{1}{\sqrt{2}}|\uparrow\rangle \otimes |\text{Aristotle} \uparrow\rangle + \frac{1}{\sqrt{2}}|\downarrow\rangle \otimes |\text{Aristotle} \downarrow\rangle$, he remains uncertain of which person in the future he is identical to. This solution is based on Lewis's account of personal identity (D. Lewis 1976, 1983). This approach is criticized based on the theory of semantics and reference by P. Lewis [11] and Tappenden [12]. In this paper, I will investigate the validity of the pre-measurement uncertainty approach to the incoherence problem and its consequences, while maintaining a more charitable position on the debate in language and semantics.

Pre-measurement uncertainty is not the only attempt to resolve the incoherence problem. Some authors favor post-measurement uncertainty to explain probability in EQM. For instance, Vaidman [13] proposes that, imagining Aristotle is blindfolded during the measurement, he would be uncertain who he is identical to after the measurement until he sees the results of the measurement. This approach is further developed by McQueen and Vaidman [14]. Tappenden [15] argues that this, combined with Sider's account of personal identity [16], explains the use of probability in EQM. Moreover, Papineau [17] and Tappenden [18] reject *Ignorance* as the foundation for understanding probability in EQM. (Instead, in a recent publication, Tappenden [19] embraces pre-measurement uncertainty. However, it bears more similarity to Tappenden's previous approach that rejects *Ignorance*, and remains to be justified whether it truly qualifies as a "pre-measurement uncertainty" approach. For this reason, I do not include Tappenden's recent approach in this paper.) Although I do not find their suggestions unproblematic, this paper will solely focus on pre-measurement uncertainty.

## 2. Personal Identity and Ontological Structure

Some attempts to understand EQM aim to distinguish different ontological structures of a *world* in order to address the debate of uncertainty. For instance, Wilson [20] argues that the mathematical structure of EQM itself does not decide between the overlapping view or divergence view. If different histories in EQM are not overlapped in the past, namely, they are quantitatively identical but numerically different in the past, EQM should be thought of in terms of divergence; either way, if they are numerically identical in the past, it should be thought of in terms of fission. Wilson [21] further claims that the mathematical structure of EQM remains neutral regarding the view of Individualism, which regards an Everett world (a branch in Saunders and Wallace's terminology) as a metaphysically possible world, or the view of Collectivism, which regards an Everett multiverse (everything described by the quantum state of the universe) as a metaphysically possible world. Following this line of thought, adopting the divergence view can avoid the problem posed by Saunders and Wallace's approach to solving the incoherence problem. This claim relies on a *deep* metaphysical understanding of EQM, namely, that there can be deep and important differences in whether there can be multiple qualitatively identical "worlds" corresponding to one state in EQM and that we should take the identity of *worlds* in EQM very seriously. However, I find this perspective misleading as it may undermine the very spirit of EQM that we do not need any additional structures or postulations of quantum mechanics. The common-sense four-dimensional world we inhabit merely

emerges from the quantum state, which is not primary in the ontology of EQM. As Wallace cites Dennett:

> Dennett's criterion: A macro-object is a pattern, and the existence of a pattern as a real thing depends on the usefulness—in particular, the explanatory power and predictive reliability—of theories which admit that pattern in their ontology [22] (p. 93).

The same applies to a *world* in EQM. The existence of a world is approximate and could be vague and indefinite in EQM [9,22–24]. Following this line, there is no deep philosophical inquiry to be made regarding the identity of physical objects in EQM, at least nothing deeper than the identity of physical objects in classic mechanics. The identity of physical objects or *worlds* is not a deep truth underlying the *prima facia* structure of EQM, as Wallace once put it in this way:

> There is a concept of transtemporal identity for patterns, but again it is only approximate. To say that a pattern $P_2$ at time $t_2$ is the same pattern as some pattern $P_1$ at time $t_1$ is to say something like "$P_2$ is causally determined largely by $P_1$ and there is a continuous sequence of gradually changing patterns between them"—but this concept will not be fundamental or exact and may sometimes break down [22] (pp. 95–96).

Consequently, the distinction between overlapped histories and divergent histories is merely a superficial artifact. If adopting the divergence view of EQM can avoid the problem that the overlapping view has in order to solve the incoherence problem, then there must be a substantial difference between understanding one branch in EQM as one world or multiple qualitatively identical but numerically different worlds. We would need to introduce additional structures (possibly only metaphysical rather than physical) to EQM if we want to find any deep differences between them. (Wallace also argues that the difference between overlapping histories and divergent histories is not meaningful for a similar reason [9] (p. 287).) I shall discuss the divergence view in Section 7. Although the divergence view may have its own problems, the aim of this paper is not to reject it. In Section 7, I shall argue that my analyses in this paper apply to the divergence view as well, and supporters of the divergence view will face the same dilemma.

Although there may be no deep ontological questions within EQM, it is still legitimate to inquire whether one *person* is identical to another within the framework of EQM. While it might be commonly agreed that personal identity supervenes the physical reality from a physicalism viewpoint, it is not *part* of our physical theories. As a result, it remains to be investigated *how* personal identity supervenes physical reality, as it allows for the development of different theories of personal identity within the framework of classic mechanics as the background. This inquiry differs from the question "divergence or not" mentioned earlier. Taking personal identity seriously does not necessarily burden the ontology of the underlying physical theory.

Personal identity, as I shall discuss in the following sections, forms the very core of pre-measurement uncertainty in EQM. I will introduce Lewis's account of personal identity in Section 3 and Saunders and Wallace's solution to the incoherence problem involving pre-measurement uncertainty in Section 4. I then will delve into P. Lewis and Tappenden's objection to Saunders and Wallace based on concerns related to reference and semantics. While maintaining a charitable perspective on the debates, I will propose another objection that there are no facts to be uncertain of in a common reading of Saunders and Wallace's proposal. In Section 5, I will present a modified view that suggests the existence of multiple qualitatively identical but numerically different mental states that supervene one physical state before the branching. The modified view can withstand the objections just mentioned. In Section 6, I further argue that this revised view cannot be consistent with physicalism and be successful in addressing the incoherence problem at the same time, unless we introduce some hidden variables into EQM. Finally, in Section 7, I discuss the "divergence

view" of EQM, which provides a concrete example that illustrates the analyses presented in Section 6.

### 3. The Lewisian Account of Personal Identity

The Lewisian account of personal identity, developed by D. Lewis [25,26], is an attempt to preserve personal identity as a definite and transitive relation despite Parfit's destructive arguments through Parfit's personal fission thought experiment [27] (pp. 245–280).

By virtue of the obvious analogy between the brain splitting case and branching in EQM, I will use the branching case in EQM to illustrate both the Parfitian account and the Lewisian account of personal identity here. In our scenario, the quantum state after branching is $\frac{1}{\sqrt{2}}|\uparrow\rangle \otimes |\text{Aristotle} \uparrow\rangle + \frac{1}{\sqrt{2}}|\downarrow\rangle \otimes |\text{Aristotle} \downarrow\rangle$. Let us denote the person represented by $|\text{Aristotle} \uparrow\rangle$ (or $|\text{Aristotle} \downarrow\rangle$) as Aristotle↑ (or Aristotle↓), and the person represented by $|\text{Aristotle } 0\rangle$ before the branching as Aristotle0.

According to Parfit, if we maintain that personal identity is a transitive and definite relation, Aristotle0 can only be identical to at most one of Aristotle↑ and Aristotle↓ since Aristotle↑ and Aristotle↓ cannot interact with each other after branching, and they are distinct agents making their separate decisions. (Here, the term "transitive" means that if person A is identical to person B, and person A is identical to person C, then person A is identical to person C. The term "definite" means it does not admit of degree, for example, we cannot say that person A is 50% identical to person B.) Consequently, Aristotle0 cannot be identical to both Aristotle↑ and Aristotle↓, as it would contradict the transitivity of personal identity. Hence, Aristotle0 is identical to only one of Aristotle↑ and Aristotle↓. If we uphold personal identity as a definite relation, given that the branching is highly symmetric, whether Aristotle0 is identical to Aristotle↑ or Aristotle↓ can only depend on some rather trivial differences between them. Parfit claims that such trivial relations cannot be of significant philosophical importance. Therefore, either there does not exist such a relation as personal identity which is definite and transitive, or such a relation is trivial and lacks significance.

The Lewisian account of personal identity seeks to preserve the definiteness and transitivity of personal identity by positing the existence of (at least) two persons both before and after branching: They coincide before branching but diverge afterward. In the case of EQM, there are *already* two persons present before branching: Aristotle0↑ and Aristotle0↓. Aristotle0↑ (or Aristotle0↓) is identical to Aristotle↑ (or Aristotle↓), but Aristotle0↑ is not identical to Aristotle0↓; hence, the definiteness and transitivity of personal identity can be preserved.

It is important to notice the original Lewisian account has a four-dimensional nature. According to Lewis's account, a person is a four-dimensional entity rather than a three-dimensional entity. The claim that Aristotle0↑ is identical to Aristotle↑ is not of *temporal identity*, but merely a trivial claim that Aristotle0↑ is identical to *itself*. As the same four-dimensional entity, Aristotle↑ is simply an alternative name for Aristotle0↑. Lewis calls the three-dimensional slice of a four-dimensional continuant as a four-dimensional person a *person-stage*, which is usually understood as a fully-present *person* in three-dimensionalism. A person, as a four-dimensional entity according to Lewis, is an aggregate of person-stages that belong to different times. ("A continuant person is an aggregate of person-stages, each one I-related to all the rest (and to itself). (It does not matter what sort of 'aggregate.' I prefer a mereological sum so that the stages are literally parts of the continuant. But a class of stages would do as well, or a sequence or ordering of stages, or a suitable function from moments or stretches of time to stages.)" [25] (p. 22)). In the scenario of this paper, there is only one person-stage before the branching and two person-stages after the branching. Since the quantum states $|\text{Aristotle0}\rangle$, $|\text{Aristotle} \uparrow\rangle$, and $|\text{Aristotle} \downarrow\rangle$ are all (approximately, of course) three-dimensional, we can use them to represent the corresponding person-stages for convenience. These three person-stages can constitute at least two (four-dimensional) persons: $\{|\text{Aristotle0}\rangle, |\text{Aristotle} \uparrow\rangle\}$ and $\{|\text{Aristotle0}\rangle, |\text{Aristotle} \downarrow\rangle\}$. (For simplicity, I have only included two typical person-stages for each

person.) The claim that there are already two persons present before branching means that, prior to the branching, the present three-dimensional person-stage $|\text{Aris}\textit{tot}\text{le}\rangle$ belongs to two four-dimensional persons. One ($\{|\text{Aristotle0}\rangle, |\text{Aristotle} \uparrow\rangle\}$) is identical to the only person who contains $|\text{Aristotle} \uparrow\rangle$, while the other is identical to the only person who contains $|\text{Aristotle} \downarrow\rangle$. (I assume that there is only one person who contains $|\text{Aristotle} \uparrow\rangle$ (or $|\text{Aristotle} \downarrow\rangle$) as its three-dimensional part for simplicity. Strictly speaking, there can be an infinite number of persons containing $|\text{Aristotle} \uparrow\rangle$ (or $|\text{Aristotle} \downarrow\rangle$) considering the possible infinite occurrences of branching in the future. However, this assumption will not affect the results in this paper.) These identity relations between the four-dimensional persons are transitive, but the identity relations between the three-dimensional persons (Lewis calls it *I-relation*, namely, two person-stages are in I-relation if, and only if, there is at least one person containing them) can be intransitive. Both the person-stages represented by $|\text{Aristotle} \uparrow\rangle$ and $|\text{Aristotle} \downarrow\rangle$ share the I-relation with $|\text{Aristotle0}\rangle$, but $|\text{Aristotle} \uparrow\rangle$ does not share the I-relation with $|\text{Aristotle} \downarrow\rangle$.

## 4. Saunders and Wallace's Lewisian Solution to the Incoherence Problem and Its Objections

Saunders and Wallace [3] utilize the Lewisian account as the foundation of pre-measurement uncertainty in EQM. Before the branching, Aristotle may have been fully aware that the quantum state after branching will be $\frac{1}{\sqrt{2}}|\uparrow\rangle \otimes |\text{Aristotle} \uparrow\rangle + \frac{1}{\sqrt{2}}|\downarrow\rangle \otimes |\text{Aristotle} \downarrow\rangle$, but he lacks knowledge of whether he is Aristotle0↑ or Aristotle0↓. As a result, he is uncertain whether he will observe the electron in state $|\uparrow\rangle$ or $|\downarrow\rangle$. There are no *internal* ways of distinguishing between Aristotle0↑ and Aristotle0↓ before the branching, for they are *physically identical up to the moment of branching*. If this is true, then there can be some *subjective uncertainty* in EQM, although the evolution of the quantum state is deterministic. Aristotle is ignorant of *who he is* before branching.

This solution is objected to by P. Lewis [11] and Tappenden [12]. (P. Lewis did not cite [3] in [11] since it was not published yet by that time. But P. Lewis did argue against a similar line of solution presented by Saunders and Wallace in [5,8])They argue that even if the Lewisian account is correct, neither Aristotle0↑ nor Aristotle0↓ could successfully refer to himself before the branching. Aristotle0↑ and Aristotle0↓ can only successfully refer to the single person-stage represented by $|\text{Aristotle0}\rangle$ before branching, which is commonly shared by all persons in this scene. Consequently, they conclude that it makes no sense to claim that Aristotle0↑ is ignorant of some indexical information about himself, as the utterance "I do not know whether I am Aristotle0↑ or Aristotle0↓" fails to express that "Aristotle0↑ does not know whether Aristotle0↑ is Aristotle0↑ or Aristotle0↓". In other words, their argument goes as follows: Before the branching, any singular terms in Aristotle0↑'s expressions cannot singularly refer to Aristotle0↑ but instead refer to all persons who supervene on $|\text{Aristotle0}\rangle$ at the same time; thus, the incoherence problem cannot be solved along this line. As P. Lewis argues:

> In particular, I cannot wonder further whether my use of the pronoun 'she' when pointing at the observer picks out she↑ or she↓; since she↑ and she↓ coincide at the moment, I am pointing at both of them [11] (p. 6). (P. Lewis's use of "she↑" and "she↓" is the same as the use of "Aristotle0↑" or "Aristotle0↓" in this paper.)

> Tappenden also objects:

> But HydraUP and HydraDOWN cannot each indexically refer to her own body via an utterance of 'This is my body' which has a single token sited in a single body-stage at time *T* prior to branching, because that single body-stage is common to the world-tube bodies of both HydraUP and HydraDOWN [12] (p. 311). (Tappenden's use of "HydraUP" and "HydraDOWN" is the same as the use of "Aristotle0↑" or "Aristotle0↓" in this paper.)

Saunders and Wallace attempt to develop a set of semantic rules where one single utterance can be paraphrased as two different propositions to address the objections [3]

(pp. 295–296). I do not want to meddle with the somewhat murky issues of language and semantics here. Whether an utterance can successfully refer is, unsurprisingly, sensitive to the context in which it is uttered and the semantic rules we apply. I shall remain neutral in the debate about semantics. Instead, I shall argue that, under some general restrictions, which I shall explicate in the following, there are no *facts* in EQM to be uncertain of. Whether our language can express our uncertainty is one thing, but whether there is *anything* to be uncertain of is another thing.

## 5. Two Versions of the Solution

In D. Lewis's original writing, the claim that there are two persons before branching is a trivial one. There are no mysterious or philosophical deep facts behind this claim that require investigation. In D. Lewis's original scene and also in Saunders and Wallace's discussions, there exists only one three-dimensional person-stage before branching. To assert that there are two persons *present* before the branching simply means that there are two different ways to combine this particular three-dimensional person-stage with other person-stages to constitute a four-dimensional person. (Tappenden [19] misconstrues Saunders and Wallace's approach as it "reject(s) the concept of splitting, which is arguably Everett's key idea". Everett is not concerned about personhood or personal identity. In [3], Saunders and Wallace do not challenge Everett's idea of split that "the observer state 'branches' into a number of different states. Each branch represents a different outcome of the measurement and the corresponding eigenstate for the object-system state. All branches exist simultaneously in the superposition after any given sequence of observations. [2] (p. 459)". Saunders and Wallace do not alter Everett's conceptual framework as a physical theory; they only introduce four-dimensionalism and an account of personal identity into EQM.) Before the branching, Aristotle's internal mental state and thinking process is single. If Aristotle can be uncertain of something, he must be unaware of some facts. When Aristotle feels uncertain whether he is Aristotle0↑ or Aristotle0↓ in his mind, there should be some facts that determine whether this thinking belongs to Aristotle0↑ or Aristotle0↓. However, it appears that this determination is merely a matter of our choice. It is Aristotle0↑ who is uncertain if we choose to combine the person-stage before the branching with some person-stages that observe the *z*-spin of the electron as up, and it is Aristotle0↓ who is uncertain if we choose to combine the person-stage before the branching with some person-stages that observe the *z*-spin of the electron as down. To put it more ironically, it is Aristotle0↑ who is uncertain if we choose that the thought which feels uncertain belongs to Aristotle0↑, and it is Aristotle0↓ who is uncertain if we choose that the thought which feels uncertain belongs to Aristotle0↓. There is something not decided here, and fairly we can say there is some kind of *indeterminacy*; however, such indeterminacy does not come from any further unknown facts, but only from a choice that remains to be made by us. This is not a kind of uncertainty.

(Saunders and Wallace propose that there are two or more thoughts of Aristotle before the branching, as they write: "If persons are continuants, we do better to attribute thoughts and utterances at t to continuants $C$ at $t$. That is, thoughts or utterances are attributed ordered pairs $\langle C, t \rangle$ or slices of persons $\langle C, S \rangle$, $S \in C$ not to temporal parts $S$. This is to apply whether or not there is branching. In the absence of branching we obtain the standard worm-theory view; in the presence of branching conclude that there are two or more thoughts or utterances expressed at t, one for each of the continuants that overlap at that time.Is it to be objected that thoughts or utterances have an irreducibly significance? We may grant the point that their tokenings are purely events—And as such, indeed, are identical—But the content of thoughts utterances is another thing altogether. On even the most timid forms of externalism, or functionalism for that matter, meanings are context-dependent. sentences produced pre-branching are likely to play different semantic each person subsequently, and likewise their component terms [3] (p. 295)." They consider *thoughts* as external entities. Their intention is to convey that there exist two or more contents within the agent's single thinking process in mind. Here I use "*thinking*" as the mental process and state of mind in

this paper. As the subsequent argument unfolds, however, it is a matter of our choice to decide the semantic content of Aristotle's thinking (according to semantic externalism, as Saunders and Wallace advocate).)

However, with just a few modifications, I will present another version of Saunders and Wallace's solution. If there is more than one three-dimensional entity that supervenes one single physical state $|\text{Aristotle}\,0\rangle$, the previous objections can be addressed. For instance, suppose that there are two three-dimensional person-stages, $(\text{Aristotle}0\uparrow)_3$ and $(\text{Aristotle}0\downarrow)_3$ before the branching, and both of them supervene $|\text{Aristotle}\,0\rangle$. Namely, there is only one singular physical body as Aristotle before branching, but there are multiple mental states, or some other three-dimensional entities, that supervene $|\text{Aristotle}\,0\rangle$. (The requirement that there is only one physical body as Aristotle before the branching can be relinquished if we introduce multiple qualitatively identical "worlds" or multiple physical states before the branching, with each mind of Aristotle situated in a distinct world. I shall discuss this approach in Sections 6 and 7. However, the claim that there are multiple mental states as Aristotle before the branching, which is more essential, shall remain unchanged. The term "three-dimensional" might be a bit perplexing when applied to a mental state. In this context, I am employing the term "three-dimensional" in a broad sense for the sake of convenience, aligning it with the terminology of three-dimensionalism and four-dimensionalism. A three-dimensional person-stage is momentary, while a four-dimensional person is not. From an eternalist perspective, one might uphold that there exists an overarching mental state for a person throughout all time, with their momentary mental states serving as partial "sub-states" of this ultimate mental state. I do not know who exactly upholds this view, but it is important to make a distinction here. In this paper, I call a mental state three-dimensional in the sense that it is momentary.) By having two or more minds that think before branching, which are *qualitatively* identical but *numerically* different, the objection presented in the previous paragraph can be resolved. Before the branching, neither thinking can tell which mental state it belongs to, as both share the same contents. But there are some further facts, though they might be unobservable in principle, that can determine which mental state it belongs to.

P. Lewis and Tappenden's objection concerning reference and semantics can also be resolved. While there is only one singular "physical" utterance, namely, only one string of voices is uttered, this utterance is reflected in two numerically different mental states. When Aristotle utters "I do not know whether I will be Aristotle↑ or Aristotle↓ after the branching", this utterance can be translated into different propositions for different minds. Hence, the pronoun "I" can refer to different entities before the branching. $(\text{Aristotle}0\uparrow)_3$ is uncertain whether $(\text{Aristotle}0\uparrow)_3$ will be Aristotle↑ or Aristotle↓, and similarly, $(\text{Aristotle}0\downarrow)_3$ is uncertain whether $(\text{Aristotle}0\downarrow)_3$ will be Aristotle↑ or Aristotle↓.

This revised solution is similar to some kind of the "Many Minds Interpretation of Quantum Mechanics" (MMI) [28–30]. MMI posits the existence of *indefinite minds* that supervene one singular physical state of ourselves. Some early advocates of MMI do not aim to address the incoherence problem via pre-measurement uncertainty; for instance, Lockwood does not offer any account of personal identity in Lockwood's MMI theory and rejects *Ignorance* as a necessary requirement. Lockwood claims that the idea of multiple minds supervening one physical state itself is consistent with physicalism. As Lockwood writes that "The assumption no more carries any dualistic implications than the conventional assumptions, which even physicalists allow themselves, about what it is like to be in such states [29] (p. 184)". However, in the next section, I shall argue that this option is inconsistent with physicalism if we intend to utilize it as a means to resolve the incoherence problem by pre-measurement uncertainty.

## 6. The Problem of Supervenience

As we have duplicated the person-stages in the previous section, the so-called "I-relation" between different person-stages is now reestablished as a definite and one-to-one relation. Adopting three-dimensionalism or four-dimensionalism will not influence the

conclusions in the following sections. For the simplicity of notations, I will use the notions in three-dimensionalism from now on. If adopting three-dimensionalism, there are already two persons, Aristotle0↑ and Aristotle0↓, before the branching or more. If adopting four-dimensionalism, the argumentation can be restored by replacing "Aristotle0↑" and "Aristotle0↓" with three-dimensional person-stages "(Aristotle0↑)$_3$" and "(Aristotle0↓)$_3$", and replacing "personal identity relation" with "I-relation". This notation shift is purely for convenience and does not imply the adoption of either the three-dimensionalism view or the four-dimensionalism view of personal identity.

I use the term '*physicalism*' to represent the view that human persons are *in essence* physical things. (Peter van Inwagen [31] (p. 225) defines *physicalism* as the thesis that "human persons are physical things". My definition is weaker as it allows some room to interpret what is "in essence" physical. These definitions, though not very precise, suffice for the purpose of my argument here.) Providing a comprehensive and elaborate definition here is both impossible and unnecessary. Instead, I present a relatively weak criterion of physicalism. According to this viewpoint, a human person is essentially a physical entity, and their personal identity can be *determined* if the physical state of the whole universe is determined and can in principle be deduced from the latter. It is fair and reasonable to demand that the following requirement be obtained and fulfilled under physicalism.

> Supervenience: The personal identity relations in a possible universe $w'$ are the same as the personal identity relations in a possible universe $w$, if $w$ and $w'$ are physically identical. (In simple terms, personal identity in a universe supervenes its physical state. In the terminology of EQM, the term "universe" refers to the entirety of physical existences described by the formulation of Quantum Mechanics. On the other hand, the term "world" is used to denote a specific branch in the universe under decoherence. Therefore, in this paper, I use the term "possible universe" instead of "possible world".)

This requirement is sufficiently lenient as it does not require that we can simply "read off" personal identity relations from the physical state. Such a requirement does not even exclude the possibility that personal identity relations supervene on physical states *nonlocally*. For example, if person A and B supervene on local physical states $|A\rangle$ and $|B\rangle$, respectively, whether A is identical to B may not be determined by the properties of $|A\rangle$ and $|B\rangle$ themselves. Donald [32] (p. 8) has suggested that a mind in MMI supervenes the entire history, which implies the non-locality of personal identity relations concerning physical states. Nevertheless, physicalism cannot be upheld if *Supervenience* is not satisfied.

The modified view presented in the previous section does not necessarily contradict physicalism. As we discussed earlier, physicalism does not necessarily require that only one mental entity can supervene on one single physical human body, as argued by Lockwood. However, to solve the incoherence problem via pre-measurement uncertainty, a specific kind of identity relation between persons before and after branching is needed. This requires more than multiple mental states to supervene one physical state.

The quantum state before the branching is represented by $\frac{1}{\sqrt{2}}(|\uparrow\rangle + |\downarrow\rangle) \otimes |\text{Aristotle } 0\rangle$. Following the discussions in the previous section, both Aristotle0↑ and Aristotle0↓ supervene on $|\text{Aristotle0}\rangle$ approximately. ($|\text{Aristotle0}\rangle$ is an *instantaneous* physical state. Here, the term "approximately" implies that, strictly speaking, Aristotle0↑ and Aristotle0↓ may supervene the physical states over a small period of time.) $|\text{Aristotle0}\rangle$ represents one single physical state and at least two numerically different mental states, which correspond to different persons (or person-stages). Various accounts can be proposed to explain how these mental states supervene the physical state. The simplest option is that they directly supervene on $|\text{Aristotle0}\rangle$ without any further fine-grained characterizations. We can suppose that Aristotle0↑ before branching is identical to Aristotle↑ after branching (and similarly Aristotle0↓ is identical to Aristotle↓), without loss of generality. This relation as personal identity is either *deterministic* or *indeterministic*. In a deterministic scenario, *which person after branching Aristotle0↑ is identical to* is fully determined by all facts (both physical and non-physical) before branching. In this case, no physical facts can fully explain how this

relation is determined. All we know about the relations among Aristotle0↑, Aristotle0↓, and |Aristotle0⟩ is the *bare* fact that both Aristotle0↑ and Aristotle0↓ supervene on |Aristotle0⟩, but there are no physical facts to distinguish Aristotle0↑ and Aristotle0↓ from their physical structures or to ground the fact that Aristotle0↑ is identical to one person supervening on one specific physical state while Aristotle0↓ is identical to another. Consequently, non-physical facts must come into play to determine the relations of those states. If, in a different universe, we have these non-physical facts different while keeping the physical state of the universe the same, we would arrive at a different result regarding whether Aristotle0↑ is identical to Aristotle↑. This, however, contradicts *Supervenience*.

If this relation is indeterministic (as suggested by Albert and Lower [28], that personal identity in EQM is *irreducibly probabilistic*), it would immediately violate *Supervenience*. The claim that it is indeterministic that Aristotle0↑ is identical to Aristotle↑ entails that in a possible universe, this proposition is false, which contradicts *Supervenience*.

The failure of the previous solution indicates the necessity of providing a more *fine-grained* account of how different persons supervene their physical states. This suggests that we should attempt to divide the state |Aristotle0⟩ into different parts in its mathematical formulation, each representing (or supervened by) a different person. For instance, we can rewrite the state before branching as follows:

$$\frac{1}{\sqrt{2}}(|\uparrow\rangle + |\downarrow\rangle) \otimes |\text{Aristotle } 0\rangle = \frac{1}{\sqrt{2}}(|\uparrow\rangle + |\downarrow\rangle) \otimes \frac{1}{2}|\text{Aristotle } 0(\uparrow)\rangle + \frac{1}{\sqrt{2}}(|\uparrow\rangle + |\downarrow\rangle) \otimes \frac{1}{2}|\text{Aristotle } 0(\downarrow)\rangle$$

where Aristotle0↑ supervenes the state |Aristotle 0(↑)⟩, and Aristotle0↓ supervenes on the state |Aristotle 0(↓)⟩. Treating these as functions over a subset of the overall direct product of configuration spaces in the formulation of Quantum Mechanics, |Aristotle 0(↑)⟩ and |Aristotle 0(↓)⟩ should have the same value due to symmetry. (The quantum state of *n* particles, known as the "wave function", is a function defined over the direct product of *n* configuration spaces of the background space manifold). In other words, Aristotle0↑ and Aristotle0↓ are qualitatively identical, so we should expect that |Aristotle 0(↑)⟩ and |Aristotle 0(↓)⟩ have the same value. One might suggest that since |Aristotle ↑⟩ and |Aristotle↓⟩ are different, |Aristotle 0(↑)⟩ and |Aristotle 0(↓)⟩ should have different values, accordingly. This proposal implies *teleology* or *fatalism*, making it hardly plausible. Suppose Aristotle does not measure the *z*-spin of an electron, but rather the sum of *z*-spins of two electrons, and the state |Aristotle 0⟩ keeps fixed; it seems that how different mental states supervene on |Aristotle 0⟩ should not be influenced by which measurement is going to be performed later. Furthermore, to distinguish |Aristotle 0(↓)⟩ and |Aristotle 0(↓)⟩ as different physical states, we ought to offer a different understanding of *what a physical state is* according to its mathematical formulation. This might require developing a new mathematical formulation of QM to differentiate them mathematically; we could envision reformulating QM as a kind of *fiber bundle* theory, where |Aristotle 0(↑)⟩ and |Aristotle 0(↓)⟩ represent different fibers upon the same element |Aristotle0⟩ in the base space, or some other alternative approach.

In Section 7, I will discuss a proposal that this can be achieved without introducing any additional mathematical structures, only through a shift of metaphysics. Following this line, it is not necessary to propose that multiple mental states supervene one physical state. Instead, they may supervene on different physical states or different "worlds". However, even if we can distinguish |Aristotle 0(↑)⟩ and |Aristotle 0(↓)⟩ based on their mathematical forms, the challenge of *Supervenience* remains. Physical facts alone cannot ground why the person supervening on |Aristotle 0(↑)⟩ is identical to the person who supervenes on |Aristotle ↑⟩ rather than |Aristotle ↓⟩, given that |Aristotle 0(↑)⟩ and |Aristotle 0(↓)⟩ have the same value. The formulation of a fiber bundle theory still lacks sufficient asymmetry to determine the relation, and the analysis presented in previous paragraphs can be equally applied here.

As Barrett [33] (pp. 185–206) suggests, giving a deterministic law of such identity mentioned above leads to some form of *hidden variable theories*. Such hidden variable theories are ad hoc if their acceptance is only for solving the issues of personal identity,

implying that we have special *connecting rules* for mental entities, but not for all physical objects. Moreover, it remains challenging to determine how such connecting rules could be. For example, if we label |Aristotle ↑⟩ with a hidden variable "↑", it could indicate a form of *fatalism* that Aristotle *must* measure the z-spin of the electron before branching. If Aristotle chooses to measure the x-spin of the electron instead, the hidden variable "↑" would hardly be effective in determining personal identity relations. I shall elaborate on this point in the next section with a particular example: the "divergence view" of EQM.

## 7. The Divergence View

Saunders [6] and Wilson [20,21] have developed the so-called "divergence view" of EQM that there are multiple qualitatively identical but numerically different *worlds* before the branching. The motivation of Saunders's proposal is to avoid the problems of Saunders and Wallace's [3] original solution to the incoherence problem, while the motivation of Wilson's proposal is probably to build a bridge between David Lewis's theory of possible world and EQM. Although Wilson claims that the choice between the divergence view and the overlapping view neutral in terms of the mathematical structure of EQM [20,21], and their proposal requires a *deep* understanding of the ontology of EQM. (In our scenario, for example, the view that there is only *one* world represented by the quantum state $\frac{1}{\sqrt{2}}(|\uparrow\rangle + |\downarrow\rangle) \otimes |\text{Aristotle } 0\rangle$, which has two different future branches, is attributed to the overlapping view.) It requires a substantial ontological difference whether $\frac{1}{\sqrt{2}}(|\uparrow\rangle + |\downarrow\rangle) \otimes |\text{Aristotle } 0\rangle$ represents one world or two qualitatively different worlds. I do not engage in the debate of whether we should accept the divergence view or the overlapping view in this paper. Instead, I argue that supposing the divergence view is correct, the discussions presented in Section 6 are still applicable to their proposal.

Saunders [6] attempts to make some room for multiple three-dimensional persons before branching by proposing that histories in EQM that share the same past *diverge* rather than *overlap*. (Saunders acknowledges to me that this is the motivation of Saunders's view on 20 October 2022 in [34].) Saunders uses an ordered pair $(\beta, |\alpha\rangle)$ to represent a *person*, where $\beta$ is what Saunders calls a momentary configuration (our |Aristotle0⟩ is an example), and $|\alpha\rangle$ is an "entire history" consisting of $\beta$ (*ibid.*, pp.191–192). In our case, there are at least two entire histories consisting of |Aristotle0⟩, whereas they consist of |Aristotle↑⟩ and |Aristotle↓⟩, respectively. I call these histories |α↑⟩ and |α↓⟩ for convenience. It seems quite natural that (|Aristotle0⟩, |α↑⟩) is identical to (|Aristotle↑⟩, |α↑⟩) and that (|Aristotle0⟩, |α↓⟩) is identical to (|Aristotle↓⟩, |α↓⟩).

Following this line, there are multiple (three-dimensional) persons before the branching, and it seems that there can be some facts to ground Aristotle's curiosity about "whether I am Aristotle0↑ or Aristotle0↓" before the branching. But, this proposal still needs to be scrutinized following the analyses in Sections 5 and 6. Again, if Aristotle is uncertain of whether he is (|Aristotle0⟩, |α↑⟩) or (|Aristotle0⟩, |α↓⟩), what facts remain unknown for Aristotle? The situation is similar to the discussion in Section 5. Once again, he is (|Aristotle0⟩, |α↑⟩) if we combine |Aristotle0⟩ with a z-spin up future, and he is (|Aristotle0⟩, |α↓⟩) if we combine |Aristotle0⟩ with a z-spin down future. This is still a matter of choice rather than a kind of uncertainty.

This rejection might be too quick, and probably the core feature of the divergence view is overlooked. The proliferation of persons is grounded in the proliferation of *worlds*. This is more explicit in Wilson's writings that:

> Then the two histories are exactly similar up to and including the penultimate projection operator, but differ on the final projection operator—they agree at all times up to $t_{n-1}$, but differ at $t_n$. The point at issue between the diverging and branching interpretations is whether the entities represented by the projection operators $\hat{P}_{\alpha_0} \ldots \hat{P}_{\alpha_{n-1}}$ in $C_{\underline{\alpha}}$ are numerically identical to the entities represented by the projection operators $\hat{P}_{\alpha'_0} \ldots \hat{P}_{\alpha'_{n-1}}$ in $C_{\underline{\alpha}'}$, or whether they are (numerically

distinct) qualitative duplicates. Numerically identical entities give us overlapping worlds; qualitative duplicates give us diverging worlds [20] (p. 73).

Here, Wilson uses symbols of consistent histories. $\hat{P}_{\alpha_0} \ldots \hat{P}_{\alpha_{n-1}}$ and $\hat{P}_{\alpha'_0} \ldots \hat{P}_{\alpha'_{n-1}}$ represent the physical reality before the branching. $C_{\underline{\alpha}}$ and $C_{\underline{\alpha'}}$ represent the complete histories that are the same before the branching. $\hat{P}_{\alpha_0} \ldots \hat{P}_{\alpha_{n-1}}$ and $\hat{P}_{\alpha'_0} \ldots \hat{P}_{\alpha'_{n-1}}$ are exactly the same with respect to mathematical formalism, and Wilson claims that they can be used to represent different ontological realities before the branching: they represent two worlds before the branching. Therefore, there can be two qualitatively identical but numerically different persons Aristotle, Aristotle0↑ or Aristotle0↓, who exist in different worlds, respectively. Aristotle0↑ will see the *z*-spin is up and the future observational result for Aristotle0↓ will be down, making it reasonable for Aristotle to be uncertain whether he is Aristotle0↑ or Aristotle0↓.

This possibility is discussed in Section 6, where it is proposed that distinguishing different physical states before the branching would require some more fine-grained mathematical structures, such as a *fiber bundle*. Wilson's approach does not require a different mathematical structure of EQM, but a different metaphysical structure of it. I do not intend to reject such metaphysical possibility here. However, we still need to address the question raised in Section 5: Is personal identity here, as a relation, deterministic or indeterministic? For simplicity, I suppose without loss of generality that Aristotle that lies in the world $\hat{P}_{\alpha_0} \ldots \hat{P}_{\alpha_{n-1}}$ is Aristotle0↑, and the Aristotle that lies in the world $\hat{P}_{\alpha'_0} \ldots \hat{P}_{\alpha'_{n-1}}$ is Aristotle0↓. Suppose that $C_{\underline{\alpha}}$ is the branch where Aristotle sees the *z*-spin is up, and $C_{\underline{\alpha'}}$ is the branch where Aristotle sees the *z*-spin is down. If the relation (personal identity) is indeterministic, it would violate *Supervenience*. One might argue that the identity of worlds across time is indeterministic, and thus *Supervenience* is preserved: In each case, the identity of Aristotle strictly follows the identity of worlds. According to this view, if the world $\hat{P}_{\alpha_0} \ldots \hat{P}_{\alpha_{n-1}}$ is identical (across time) to the world where Aristotle sees the *z*-spin is up, then Aristotle0↑ is identical to Aristotle↑, not Aristotle↓. However, this introduces indeterminacy of the identity between worlds. Supporters of the divergence view cannot deny that this is an additional character that originally EQM did not have: indeterminacy.

If such a relation (personal identity) is deterministic, it must be grounded in some physical facts that establish a deterministic connection between worlds (or the identity of worlds across time, in other words). In this case, $\hat{P}_{\alpha_0} \ldots \hat{P}_{\alpha_{n-1}}$ is connected to the (future) world where Aristotle sees the *z*-spin is up, and $\hat{P}_{\alpha'_0} \ldots \hat{P}_{\alpha'_{n-1}}$ is connected to the world where Aristotle sees the *z*-spin is down after the branching. This introduces *hidden variables* into EQM: Each qualitatively identical world before the branching is labeled with a hidden variable to determine its future successor. This notion is termed "many-threads theory" by Barrett, as Barrett explains that:

> That is, if one includes the global wave function in the state description of the worlds, then each world might be thought of as being described by a particular hidden-variable theory, where the preferred basis selects the always determinate physical quantity (the hidden variable), the local state of each world at a time gives the value of this quantity in that world, and the connection rule (together with the linear dynamics) determines, in so far as it is determined, how the quantity evolves in each world: A many-threads theory is ultimately just a hidden-variable theory where one simultaneously considers all physically possible worlds [33] (pp. 183–184). (It seems to me that Wilson does not pay much attention to Barrett's alarm in Wilson's writings. Wilson only cites Barrett once in [35] without mentioning this point. I am grateful to Shan Gao who reminds me of Barrett's writing.)

Wilson [20] (p. 69) does acknowledge that "'Many worlds' or 'many minds' theories which posited additional fundamental structure would not be worth the price." It is not necessary to introduce hidden variables into the divergent view in discussing the ontology of EQM, so Wilson does not need to be concerned with that in [20]. However, this is indeed a problem if we want to solve the incoherence problem of EQM via pre-measurement uncertainty. If we want to avoid complicating EQM as a physical theory, we have to introduce a connection rule to determine the successor of different qualitatively identical persons before the branching, which leads to a violation of *Supervenience*. The introduction of the divergence view here serves as an illustration of the various possibilities discussed in Section 6.

### 8. Conclusions

So far, I have examined approaches to solve the incoherence problem of EQM via pre-measurement uncertainty. Through a comprehensive analysis of Saunders and Wallace's solution based on David Lewis's account of personal identity, I have argued that the pre-measurement solution to the incoherence problem cannot be successful if only one mental state supervenes each observer's physical state in EQM. This need not prove fatal to the pre-measurement approach if there can be multiple qualitatively identical but numerically different mental states supervening each observer's physical state. However, the latter approach can only be successful while violating principles of physicalism. I use the "divergence view" of EQM as an example to illustrate my argumentation. As I have argued in Section 6, this brings us back to the old problems of EQM. Either we need to accept a form of "Many Worlds Theory" by introducing hidden variables into EQM, or we have to develop a kind of "Many Minds Theory" that violates principles of physicalism. My analysis in this paper is impartial regarding the adoption of three-dimensionalism or four-dimensionalism, as well as the overlapping view or the divergence view of EQM. My argument also circumvents the debates on the theory of semantics and reference, upon which previous criticisms of Saunders and Wallace's proposal have rested.

An anonymous reviewer reminds me that "at the Tel Aviv conference [34], several participants argued for the introduction of hidden variables to Many Worlds theory and also for the introduction of objective probability, also distinctly non-Everettian." Indeed, this remains a possibility for EQM. However, after introducing non-Everettian elements into EQM, it still needs to be justified why EQM should be preferred over other interpretations of quantum mechanics. This may be encouraging news for those who favor post-measurement uncertainty or probability without uncertainty in EQM, though I believe that those solutions have their own problems. Discussing these options goes beyond the scope of this paper. For those who are reluctant to complicate our physical theories by adding non-Everettian elements to EQM, embracing non-physicalism remains an option. In this sense, I believe that the Many Minds Interpretation (MMI) deserves more attention than it has received in the literature today.

**Funding:** This research received no external funding.

**Data Availability Statement:** Not applicable.

**Acknowledgments:** I am grateful to Arthur Schipper (Peking University), Sebastian Sunday Grève (Peking University), Chunling Yan (Institute of Philosophy, Chinese Academy of Sciences) and Shan Gao (Shanxi University) for comments and suggestions on previous versions of this paper.

**Conflicts of Interest:** The author declares no conflict of interest.

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
