# Peer review of "Personal Identity and Uncertainty in the Everett Interpretation of Quantum Mechanics"

_quantumrep, doi:10.3390/quantum5030038_

Round 1
Reviewer 1 Report (Previous Reviewer 1)
See attached file

Minor changes needed which I’ve specified
Author Response
See attached file.

Reviewer 2 Report (New Reviewer)
Review of Personal Identity and Uncertainty in the Everett Interpretation of Quantum Mechanics
The paper tries to address a matter of Quantum Physics from a philosophical viewpoint. There is no objection to that being done. Quantum Physics was beset with philosophical problems since its inception, as pointed out by Max Jammer in his book, The Philosophy of Quantum Mechanics. However, the discussion always devolved on matters of hard Physics that could, at least in principle, be decided by physical experiments. In this case, however, it has entered into the domain of philosophy for which there can be no physical experiment. The author is following on a sequence of papers by philosophers published in philosophical journals, but the author wants to publish this attempt in a journal of Physics. More specifically, it concerns the Everett Many World Interpretation.
Though the discussion of the philosophers is perfectly sound in its context, it misses the whole point of the Everett interpretation. While Quantum Theory does not entail any paradoxes when dealing with an ensemble of quantum entities, problems arise when it tries to deal with single entities. This interpretation addressed the Schrödinger cat paradox, which appears to entail a cat being both dead and alive simultaneously till it is actually observed to be one or the other. Everett replaces the ensemble of quantum entities by an ensemble of universes, so that in each universe the cat is either dead or alive but in the superposition of universes, the “collective cat” is both. On replacing the cat by a person, the question of the identity of the person becomes problematic. Is it the same person in all the universes, or is the “collective person” just a fiction?
From the physical point of view the cat is a red herring introduced for dramatic effect. What is relevant is the quantum entity each universe. Since we deal with identical quantum particles in a single universe as a matter of course, there is no problem with “the identity” of any. However, because of our personal identification with people, for them it becomes a serious problem. No physical experiment would be able to address the subjective experience of being a single individual. For the physical viewpoint, in his book, the Fabric of Reality, David Deutsch points out that a quantum measurement leads to some universes splitting from the others, rather than for a split occurring within one Universe. The philosophers discussing this point have not been able to get this point through their skulls. This lack of understanding of Everett’s interpretation leads the author (and the other philosophers) to unnecessary mental contortions. The ignorance that introduces probability into the discussion is as to which universe has which outcome, and there is really no need for the rest of the paper for the physical discussion. That can never begin to address the philosophical issue of the uniqueness of the person’s identity, or its lack.
To summarize the point, the philosophical discussion does not address the physical issue of probability entering into a deterministic discussion, just like this physical discussion would not address the philosophical matter of the uniqueness of the identity of individuals. As such, a paper on the philosophical discussion would be out of place in a journal of Physics and the physical discussion would be irrelevant for the philosophical issue, and would not be appropriate for a journal of Philosophy.
For this paper, the confusion of the two issues would make it useless for either type of journal. It is badly put together, too wordy for a Physics paper and missing the key philosophical issue for a Philosophy paper.
It also has issues of errors of English, formatting and using an acronym at the start of the Abstract without bothering to define it.
Author Response
I have corrected errors of English and inconsistency of formats in this paper. I have also defined the acronym at the start of the Abstract. Thank you for your precious comment.

Reviewer 3 Report (New Reviewer)
The manuscript presents an interesting idea. However, the authors need to address several points in order to make the manuscript suitable for publication. For example, some sentences seem confusing since some of them start or end in first person whereas the body of the text does not follow the same style. The custom writing style is in the plural ("we"). The authors should correct this style error, throughout the manuscript. The authors need to proofread the manuscript since several ideas seem to be inserted and do not seem integrated within the manuscript, cutting the flow of the general reading between neighboring sentences and leading to a misunderstanding of paragraphs and their main ideas.
1. All initials and acronyms should be explicitly described, regardless of how evident they seem in the field. Each letter in the acronym should be explained using parentheses showing their meaning EQM (…).
2. The first four sentences up to line 8 need to be integrated into a single idea to give more impact.
3. In general, despite there being only one author, the manuscript should be written in the plural, for example, replacing “I discuss’’ with “We discuss” throughout the manuscript.
4. A reference should be included on page 1 line 19, at the end of “Born Rule”.
5. The sentences in lines 20 and 21 should be modified to keep the idea. The period separating lines 22 and 23 should be removed in order to join those paragraphs.
6. In line 4 (as well as in important sections or paragraphs where it is also used) the element “Alexander” should be properly cited since it is vital for the manuscript.
7. Lines 35 to 38 could be removed by adding a reference just before the colon in line 40, right after “… physical theory”.
8. The equations in lines 25-26 and 28-29 should be moved outside the text and need to be integrated into the format of the equation to be cited in the text, avoiding being written repeatedly.
9. The introduction as a preamble is too long. Due to its title “Introduction: The Incoherence Problem” it can be separated into a preamble describing the work that will be developed in the manuscript, resembling the text from line 47. Such a preamble can be moved to a subsection or integrated into a section regarding The Incoherence Problem.
10. All sentences “in my paper” should be replaced with “in this manuscript”, or “in this work” or using plural or using a proper reference in case the authors refer to another work.
11. A preamble is missing in the figure captions. The presented figures need to be explained. Moreover, there is also necessary to refer to Figure 1.
12. Lines 171-173 should be integrated into a single sentence to give more relevance and impact to the idea of “a person is a 4-dimensional … ”.
13. There is a difference between a Ket and a Bra in the Dirac notation and between the symbols larger and smaller than, visually. The manuscript should be modified following the proper symbology.
14. Lines 375-376 should be written as equations.
15. After reading section 4, the reader might expect to recognize the results in a precise way given the preamble in previous sections. However, it is not the case. Hence, I recommend including a “Results” section where such points are indicated, giving more impact to the work and its main idea. In this section the preamble elements should be used such as the incoherent problem, the EQM elements, a person dimensionality, Saunders and Wallace’s Lewisian solution, etc., to illustrate the secondary goals (which should also be stressed in the abstract and conclusions) concisely along with the procedure to obtain them, and finally, the main objective, which is mentioned in the manuscript, obtained from the secondary objectives.
16. The abstract and conclusions need to be re-written, for example, in the abstract, the first sentence does not contribute to the main objective since it is not connected with any sentence, moving the focus away from the manuscript's relevance. Moreover, the significative contribution of the work and novelty regarding other works should be clearly stated (which is shown with a larger impact if written in plural). In the conclusions, the use of references is encouraged and the mentioned conference should be properly cited since it seems an informal talk. Such mention should be in the results section to highlight the manuscript's objective and hence, cite it only in the conclusion. While comparisons are valid in the conclusions, I encourage the authors to write them such as the sentences synthesis impacts on the manuscript results. It is also valid to create a section entitled “Discussions” or “Results and discussions” to compare and mention such comparisons to increase the manuscript's impact regarding similar works.
17. I recommend updating the references since there are only references to “Wilson, Alastair” which seem up-to-date, and in the EQM context, the manuscript seems to be of low relevance which is not the case, due to the presented idea and title.
The english quality seems fine, with no major issues. Only the typical recommendatios regarding the common unnoticed errors at a first glance. A mandatory modification that should be implemented is changing the writing to plural first person.
Round 2
Reviewer 2 Report (New Reviewer)
I have more or less randomly filled in the fields for this revision. The reason is that while the English has been improved, the author refuses to, or is unable to, understand the main point made by me. There is an ensemble of Universes in what he wants to denote by E. In his terms, the ignorance is of which of the universes the observer is in. He ignores the ensemble and so runs into the same trouble as he did before and the philosophers he quotes had done. If he once accepts the ensemble of universes in a single Multiverse (as Deutsch calls it), he problems he is "solving" disappear. He must address that issue.
Further, tracking all the changes is too tedious. If he still has something to say, I would like to see it as a new manuscript not showing tracking.
None at this stage.
Author Response
I am very sorry that you feel tedious to read the manuscript with trackings. In fact I had submitted two versions: one with trackings and one without trackings. Maybe the editor did not show you the manuscript without trackings.
I acknowledge that I did not fully understand your main point, and I still have difficulty to understand the point you are making. In the sentence "There is an ensemble of Universes in what he wants to denote by E", it seems that the reviewer uses "he" to refer to me, but I did not use the symbol "E" in this paper. The term "an ensemble of Universes" is unclear. In statistical mechanics, the notion "ensemble" is abstract; The states in an ensemble cannot be all real. If the reviewer uses "ensemble" in this sense, it bears more familiarity to Wilson's idea of "the divergence view", which I have already discussed. If "an ensemble of Universes" means many universes described by the mathematical form of Quantum Mechanics which are equally real, this line understanding is not different to David Wallace's.
In the previous report, the reviewer addresses that "David Deutsch points out that a quantum measurement leads to some universes splitting from the others, rather than for a split occurring within one Universe." I have discussed the divergence view and the overlapping view in this paper. According to the divergence view, there is one universe (as ours) splitting from the others. According to the overlapping view, one universe splits into multiple universes. In neither case I accepted the notion that "a split occurring within one Universe". It seems that the idea that "the ensemble of universes in a single Multiverse" is not a different proposal which I have not discussed in this paper.
Reviewer 3 Report (New Reviewer)
Lines 421 and 422 must be aligned
no comments
Author Response
I carefully reviewed the formatting of the entire text, including the formula mentioned in lines 421-422.
This manuscript is a resubmission of an earlier submission. The following is a list of the peer review reports and author responses from that submission.
Round 1
Reviewer 1 Report
Please see uploaded file

Reviewer 2 Report
find it totally unsuitable for publication in a journal devoted to publishing serious research in physics. One basic point is that the author relies fairly heavily on the notion that "Alice", a macroscopic object is described by a wave-function in the usual sense. This is of course wrong: any macroscopic object is described in quantum mechanics by a density matrix, which eliminates all question related to a global phase that is absent in density matrix. This ignorance is -unfortunately- fairly common in philosophical papers discussing quantum mechanics paradoxes etc. By the way, Alice is likely a lady, so should referred o as "she", not as "he" as done in this paper.
Author Response
Thanks to the reviewer for reading my manuscript. As to the first question, I do not see what globe phase has anything to do here. And I am following the way to represent a human person by wave function as writers whom I cite in my paper do.
As to the last question, I used the pronoun “he” because I see “he” as a gender-neutral pronoun (we use “he” to call God and each person of God in Christianity, while using “she” to call God is too anthropomorphist, and using “they” to call God is to confuse Dyophysitism with Nestorius’s view that there are two persons as Iesus Christ). It is obviously confusing to use “they” here because how many persons of Alice there are is the core question in my paper, and grammatically I think “they” is not a singular pronoun. Some people might disagree that “he” is a gender-neutral pronoun, due to my personal preference to insist “he” I have replaced all “Alice” by “Alexander” in my paper.